# A male-drive female-sterile system for the self-limited control of the malaria mosquito *Anopheles gambiae*

Anna Strampelli, Katie Willis ⓘ , Hannah Robyn Gulliford, Matthew Gribble, Barbara Fasulo, Austin Burt ⓘ , Andrea Crisanti ⓘ & Federica Bernardini ⓘ ✉

Despite great leaps forward in preventing and treating malaria, several challenges, including insecticide resistance, have hindered progress in fighting the disease. Thus, there is a pressing need for new tools to control malaria, including the use of genetically modified mosquitoes (GMMs) in the field. Various genetic strategies for vector control are currently explored, ranging from self-sustaining GMMs with unrestricted geographic and temporal spread to self-limiting alternatives. Here, we describe a self-limiting gene drive strategy called Male Drive Female Sterile (MDFS) targeting *Anopheles gambiae*, a major malaria vector. The MDFS genetic construct causes dominant sterility in females, while transgenic males remain fertile, allowing them to transmit the female sterility trait at super-Mendelian rates. Laboratory studies show that repeated releases of MDFS can lead to elimination of caged mosquito populations. Based on these findings, modelling suggests MDFS could be a highly effective and self-limiting strategy for suppressing wild malaria mosquito populations.

Despite ongoing global efforts, malaria remains a devastating disease that claimed over 600,000 lives and affected ~249 million people in 2022[1]. *Anopheles gambiae* is one of the most effective and efficient vectors of the disease and is endemic in sub-Saharan Africa, where about 96% of malaria-related deaths occur[1,2].

Insecticide-based vector control has long been a crucial element in the fight against malaria and remains essential in current prevention efforts. However, global progress in the field is threatened by numerous challenges, most notably the emergence of insecticide resistance[3]. As such, the research and development of new tools have been identified as a top priority in the malaria control agenda.

One promising approach is genetic vector control, which involves intentionally introducing specific genetic traits into a target mosquito population. This approach involves two main, non-mutually exclusive strategies: population suppression and population replacement. Population suppression strategies aim to reduce mosquito vector populations to levels that are insufficient for sustaining the transmission of the malaria parasite. In contrast, replacement strategies focus on modifying the target population to become resistant to the parasite's development or transmission[4].

Genetic vector control strategies can be further categorised based on the extent to which the introduced genetic traits and their effects spread and persist within and beyond the target population.

The first category is referred to as "self-limiting." This approach requires repeated and substantial releases of genetically modified mosquitoes (GMMs) to achieve the desired effect in the target population. The impact is expected to be temporary and contained both in time and space, allowing the population to recover once the releases are halted.

Two classical self-limiting systems for population suppression include the release of insects carrying a dominant lethal (RIDL), where males carry a gene that causes lethality in their progeny, and its female-specific variation, fsRIDL, where the lethality is restricted to females, leaving males unaffected and able to transmit the 'lethal' gene to subsequent generations[5]. RIDL and fsRIDL systems have been developed in various insect species that are significant for both agriculture and public health[6–12].

Department of Life Sciences, Imperial College London, London, UK. ✉e-mail: f.bernardini11@imperial.ac.uk

Another self-limiting strategy, "X-shredder", leads to population suppression by biasing the sex ratio toward males. This system was first developed in *An. gambiae* by using the spermatogenesis-specific *β2-tubulin* promoter to express endonucleases, such as I-PpoI or Cas9, able to recognise and cleave a conserved sequence within the ribosomal DNA repeats, exclusively located on the X-chromosome[13–15]. As a result, the X-bearing sperm of transgenic males are damaged, leaving predominantly Y-bearing sperm to fertilise the eggs, and leading to a male bias of approximately 95% in their offspring. Repeated releases of males hemizygous for the *I-PpoI* transgene have successfully led to the elimination of mosquito populations in small cages[14] and significantly reduced larger populations in large cages[16]. The X-shredder system has since been implemented in other mosquito species[17] and in fly species[18,19].

A second category of genetic vector control strategies, termed 'self-sustaining', can, in theory, spread indefinitely through a target population from release frequencies as low as 1% relative to the target population's size within its natural boundaries[4,20,21].

Within this category, CRISPR-homing gene drives are the most studied and advanced in development and have been generated in various model and vector species for both population suppression[22,23] and population replacement[24–30].

Typically, such gene drives consist of two effectors linked in a single construct: a Cas9 expressed under a germline-specific promoter and a guide RNA (gRNA) targeting a haplo-sufficient gene. In many pest and vector insect species, such as malaria mosquitoes, females play a crucial role in reproductive capacity and are responsible for transmitting pathogens. Therefore, suppression strategies often target genes that impact the fertility or viability of female insects and allow males to spread the genetic construct, imposing a higher load on the target population[31–33]. Importantly, the gene drive construct is inserted within the gene it targets. The expression of Cas9 in the germline

promotes the cleavage of the wild-type allele located on the homologous chromosome, converting it into a gene drive allele through homology-directed repair (HDR). This mechanism allows the gene drive to bias its inheritance, with transmission rates exceeding Mendelian expectations (greater than 50%). Notably, hemizygous gene drive females remain viable and fertile and contribute to the gene drive's exponential spread through the target population until the progressive accumulation of homozygous, unviable or sterile females causes its collapse.

The most successful iteration of this strategy is the CRISPR-homing gene drive targeting *doublesex*, a highly conserved gene that regulates sex differentiation in insects[23,34,35]. In *An. gambiae*, the *doublesex* transcript is alternatively spliced into a female-specific (*dsxF*) or male-specific (*dsxM*) isoform, differing in the retention of exon 5 in females and its excision in males[36]. A CRISPR-homing gene drive targeting the intron 4-exon 5 boundary in the *doublesex* gene results in the loss of functional *dsxF* transcript and, in homozygous females, leads to an intersex phenotype and full sterility[23]. In caged experiments, the single release of hemizygous gene drive males at frequencies as low as 12.5% eliminated populations in both small[23] and large cages[37], underlining the high potential of this technology for vector control.

The potential for self-sustaining systems to spread across large geographical areas upon their release makes these strategies highly efficient and cost-effective for large-scale implementation. However, although laboratory results have shown promise, self-sustaining strategies have not yet been implemented in real-world settings, and ongoing efforts are focused on regulating potential field releases[38–40].

In contrast, there are a growing number of examples of self-limiting strategies being implemented in the field or approved for future use[12,41,42]. However, the need for multiple releases and the rearing of large volumes of mosquitoes are both impractical and costly, limiting the wide-scale use of these systems, particularly in vast rural areas of sub-Saharan Africa, where they are most needed.

In this scenario, much attention has been given in recent years to genetic strategies based on the CRISPR technology and including a driving component, yet self-limiting and less invasive. These strategies are expected to achieve appreciable population suppression locally or regionally with relatively low release frequencies, making them valid and efficient tools for malaria control[43].

Examples of such strategies are increasingly being theorised, developed and tested. These include Y-linked editors, whereby a Cas9/gRNA construct located on the male-specific Y chromosome makes dominant edits to female-specific genes[44,45] and split drives, in which the components of a CRISPR-based gene drive, Cas9 and gRNA, are unlinked and the driving mechanisms only become active when the two elements are both present in the same individual. Most of these split drive systems are being developed for population replacement[46–55], with only a few aimed at population suppression[56–58].

Another system, fs-RIDL-drive, consists of an allele that causes dominant lethality or sterility in females, similar to fsRIDL, and is transmitted at super-Mendelian rates by males, akin to gene drive[22,52,58,59]. As such, this strategy can be seen both as a potentiated version of fsRIDL[5,60] or as a temporally and spatially restricted version of a homing gene drive[22,23].

The differences in the dynamics of spread and efficacy of the self-limiting strategies discussed are illustrated in Fig. 1, which depicts time-series simulations of releases using a simple deterministic model. Burt and Deredec predicted the fs-RIDL-drive system to be more efficient than the other self-limiting strategies, including RIDL, fsRIDL and the X-shredder, yet remaining non-invasive[44]. Additionally, the authors found that coupling fs-RIDL-drive to the X-shredder would lower the number of male releases needed to achieve a set level of suppression by more than an order of magnitude, thus greatly improving its efficacy (Fig. 1).

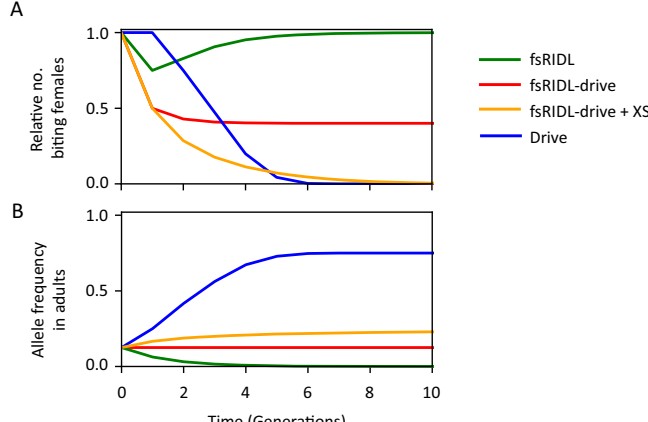

**Fig. 1 | Time course simulations following a single release of males hemyzygous for different genetic constructs at 100% of the initial male population show the relative number of biting females. A** and the allele frequency (**B**). Strategies include *i)* fs-RIDL (green), where males are released carrying a single copy of a dominant lethal gene which causes lethality in females, *ii)* fs-RIDL-drive (red), where the same dominant lethal allele is engineered to drive via homing in males, *iii)* fs-RIDL-drive + XS (orange) where the latter is released alongside a second unlinked genetic construct which causes males to produce only Y-bearing sperm, and finally *iv)* Drive (blue) where males are released carrying a single copy of a recessive lethal gene affecting only females which is capable of driving via homing in both sexes. All simulations are produced using a deterministic model of a single panmictic population with two sexes and two life stages, with density-dependent mortality occurring at the juvenile stage. All strategies are modelled with idealised parameters assuming fitness effects cause lethality after density-dependent mortality, which is equivalent to disrupting a gene which causes female sterility and prevents biting.

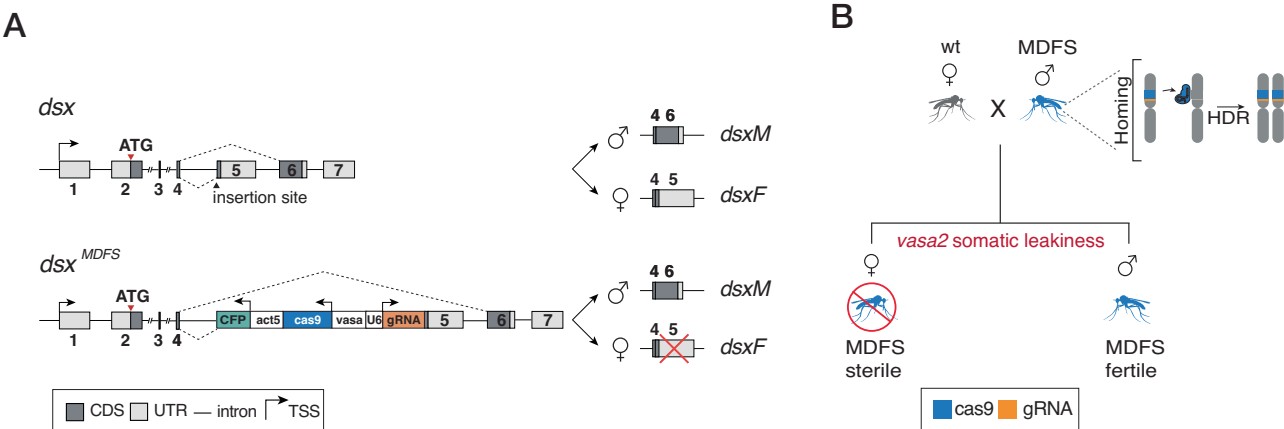

**Fig. 2 | Schematic representation of the MDFS construct insertion at the *doublesex (dsx)* locus and expected progeny outcomes from carrier males. A Top**, schematic organisation of the *dsx* gene. Differences between the male- and the female-specific transcripts are highlighted. The black arrowhead represents the insertion site of the MDFS construct. **Bottom**, The MDFS allele is produced by φC31 RMCE using the *dsx⁻* knock-in line (Kyrou et al.), which contains an eGFP φC31 acceptor construct (not shown in the figure), inserted into the *dsx* gene at the boundary of intron 4 – exon 5 (indicated by the insertion site arrow). The genetic construct contains an *actinSc::eCFP* fluorescent marker, a *Cas9* driven by the germline *vasa2* promoter and a *U6::gRNA^dsxF* cassette. The construct insertion hinders functional *dsxF* transcript production while leaving the *dsxM* unaffected.

Dashed lines represent splicing events. Non-coding regions (UTR) are shaded in light grey, coding regions (CDS) are shaded in dark grey, introns are represented by black lines and are not in scale. Bent arrows symbolise transcription start sites (TSSs/promoters). **B** Genetic cross between wild-type *Anopheles* females and MDFS males. In the male germline, Cas9 activity leads to the homing of the MDFS transgene via the HDR system and to the super-Mendelian inheritance of the construct. In the offspring, the leaky activity of the *vasa2* promoter in somatic tissues leads to mutagenesis at the *dsxF* locus. This results in female sterility due to non-functional *dsxF* alleles (the transgene insertion disrupts one allele, and the other has indels induced by the leaky activity of the Cas9). Figure 2 was adapted from: iStock.com/LCOSMO.

Here, we develop and test an fs-RIDL-drive strategy to target the malaria mosquito *An. gambiae*, which we term Male-Drive Female-Sterile (MDFS). We show that males carrying the MDFS allele, whether alone or combined with an X-shredder system, are fertile and transmit the MDFS allele at super-Mendelian rates, while females exhibit dominant sterility. Furthermore, we demonstrate that repeated releases of MDFS males can eliminate caged mosquito populations in a self-limiting manner. Together with our population genetics model, these results confirm the potential of MDFS as a non-invasive and effective strategy for population suppression.

## Results

### Design and generation of an MDFS system targeting *doublesex*

To generate an MDFS system, we designed a genetic construct containing an eCFP fluorescent marker, a Cas9 endonuclease under the expression of the germline *vasa2* promoter[60] and a *U6*-driven gRNA constitutively expressed targeting the female-specific exon 5 of the *doublesex* gene, which causes recessive female sterility when disrupted[23,36] (Fig. 2A).

We selected the *vasa2* promoter because it is expressed in the germline of both sexes and exhibits somatic leakiness[22,29,60–62]. We hypothesised that in hemizygote individuals carrying the MDFS construct, the *vasa2* promoter would cause the expression of Cas9 in somatic cells. In these cells, disruption of the *doublesex* exon 5 would result in conversion into a null allele, leading to female sterility (i.e., MDFS would exert a dominant effect)[63–66]. On the other hand, MDFS males would be unaffected, and they would transmit the allele to a super-Mendelian fraction of their progeny by virtue of the germline expression of Cas9, which promotes the homing of the MDFS allele (Fig. 2B).

We generated an MDFS strain by performing embryo microinjections of the MDFS construct into a mosquito strain containing two *attP* docking sites at the *doublesex* intron4-exon5 boundary. Integration of the cassette at the target locus was achieved via recombinase-mediated cassette exchange (RMCE), following embryo injection with a plasmid containing a source of φC31 integrase[23] (Supplementary Fig. 1A and Fig. 2A). G1 transformants were isolated, and a subsequent

molecular investigation confirmed the successful recombinase-mediated cassette exchange with the target locus (Supplementary Fig. 1B).

### MDFS females display external and internal abnormalities and are fully sterile

An MDFS strategy should cause dominant female sterility, such that hemizygous females constitute a "dead-end" and do not contribute to the construct's spread, unlike self-sustaining systems in which both sexes transmit the transgene when they carry it in hemizygosis.

Following our initial observations of the G1 transformants (see Supplementary Table 1), none of which displayed a female phenotype in adulthood, we further examined the MDFS strain we established from one of the G1 males. We found that all the positive pupae for the eCFP marker displayed male-like features, a characteristic also exhibited by females with a knock-out of *dsxF*, previously described as 'intersex'[23].

At adulthood, the sexual dimorphism between MDFS males and genetic females became more apparent, and a total of 41 adult females were selected for further investigation of their phenotype. Dissection analysis revealed external and internal morphological and anatomical abnormalities reminiscent of those observed in Kyrou et al.[23]. Externally, all MDFS females had claspers, a feature typically displayed only by males: the vast majority were dorsally oriented (n = 38/41), with a 180 degrees rotation when compared to the abdominally oriented claspers of mature males, and a small minority were deformed (n = 3/41), for example, oriented in opposite directions (Fig. 3A–D). Furthermore, MDFS females displayed heads with club-shaped maxillary palps and more extended, hairier antennae compared to those of wild-type females, resembling male features (Fig. 3E–G).

Internally, most MDFS females had ovaries (n = 37/41), though they were largely underdeveloped (Figs. 3H and 3I). Only one female from those examined had a spermatheca, which appeared significantly smaller than that found in wild-type females. Additionally, male accessory glands (MAGs) were present in all females, though in a small subset, they seemed less developed (e.g., there was only one or both were lighter in colour) (n = 8/41) (Fig. 3J).

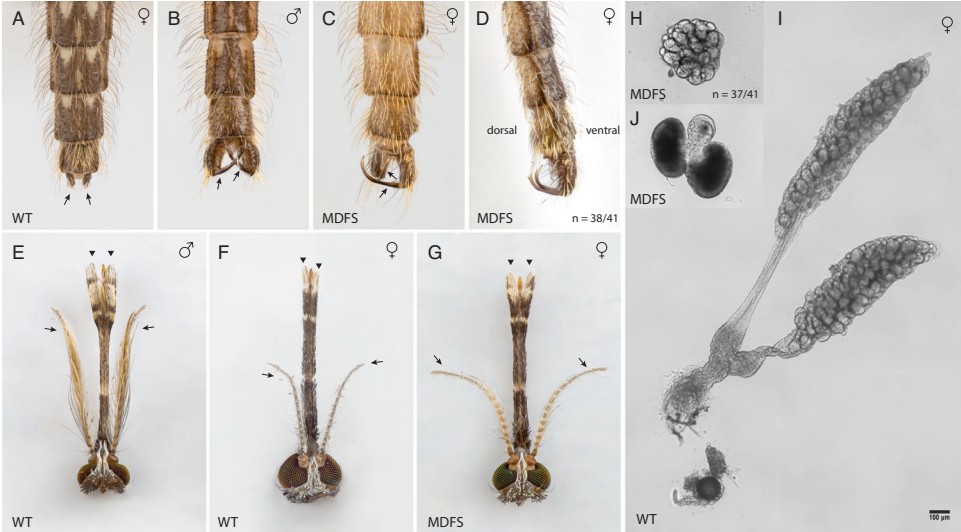

**Fig. 3 | Morphological features of MDFS adult individuals. A–D** The last four abdominal segments of an adult wild-type female (**A**), an adult wild-type male (**B**), and MDFS females (**C, D**). The last abdominal segment of wild-type females consists of a pair of finger-like structures called cerci (**A**, arrows). Instead, males exhibit claspers (**B**, arrows), a pincer-like structure to grasp the female during copulation. MDFS females have claspers (*n* = 41/41) resembling those in wild-type males (**C**). However, the claspers are usually rotated 180° towards the dorsal side (*n* = 38/41) (**D**). **E** Wild-type adult male head showing large plumose antennae with long hairs (arrows) and long club-shaped maxillary palps (arrowheads). **F** Wild-type adult female head with shorter, less hairy antennae (arrows) and slim maxillary palps (arrowheads). **G** MDFS female head with club-shaped maxillary palps (arrowheads) and more extended, hairier antennae (arrows). **I** Ovaries and spermatheca dissected from a sugar-fed wild-type female. The picture shows the difference in size between the wild-type and a single ovary extracted from one MDFS female (*n* = 37/41) (**H**). MDFS ovaries are always associated with male accessory glands (MAGs) (*n* = 41/41) (**J**)−scale bar 100 μm.

When given the chance to blood-feed, the MDFS females could not do so. As a result, mating 50 MDFS females with an equal number of wild-type males produced no progeny, with no eggs laid. In contrast, the wild-type genetic cross used as a control produced a total of 3,969 eggs. Furthermore, monitoring the MDFS strain through the rearing practices in our laboratory for over 50 generations has confirmed an intersex phenotype for all the MDFS females examined, indicating that the penetrance of the dominant sterile phenotype, by virtue of the *vasa2* promoter, has remained stable over time (Supplementary Table 2).

Finally, we collected MDFS individuals at different developmental stages (larvae, pupae, and adults), and we performed pooled amplicon sequencing at the wild-type *dsxF* allele (i.e., the target site of the Cas9 on the homologous allele to the one in which the MDFS is inserted). This analysis revealed extensive cleavage at the *dsxF* allele, which increased from 32.4% at the L1 stage (both sexes pooled) to an average of 85.3% in adult females and 80.2% in adult males (Supplementary Fig. 2A). All samples analysed exhibited a wide range of different repair outcomes (>20), mainly in the form of small insertions and deletions centred around the cut site (Supplementary Fig. 2B), supporting the hypothesis that extensive somatic cleavage driven by the *vasa2* promoter is responsible for the MDFS phenotype observed.

### MDFS males are fertile and transmit the MDFS allele at super-Mendelian rates

A second requirement for an MDFS strategy to efficiently spread is that male individuals must be fertile and transmit the MDFS allele at super-Mendelian rates. Additionally, previous mathematical modelling data indicate that the MDFS strategy could be enhanced by coupling it with a sex distorter, such as an X-shredder system[44]. In this case, males carrying both the MDFS and X-shredder alleles should be fertile, and, in addition to transmitting the MDFS allele at super-Mendelian rates, they would also produce a male-biased progeny (Fig. 4A). This unique combination of traits allows the MDFS allele to transiently drive in the population, as it would be passed on by a greater than 50% portion of the progeny (i.e., the males).

We first compared the fertility of males carrying the MDFS allele and males carrying the MDFS and an unlinked X-shredder allele to that of wild-type controls. In addition, we measured the inheritance rate of both alleles (MDFS and X-shredder) in their progeny.

To generate MDFS;X-shredder males, genetic crosses were set up between MDFS males and females of the previously characterised PMB1 strain. This strain carries the I-PpoI[I24L] endonuclease under the spermatogenesis-specific *β2-tubulin* promoter (making it inactive in females) and an RFP fluorescent marker under the *3xP3* promoter integrated into the centromeric region of the 2 R chromosome[14,67].

The fertility assay revealed that MDFS males do not exhibit fertility costs in laboratory conditions compared to their wild-type counterparts and that adding the X-shredder allele to their genetic background does not negatively impact fertility. Specifically, the average larval output of MDFS males (68.6 ± 5.6, *n* = 38) and MDFS;X-shredder males (64.0 ± 4.7, *n* = 38) were not statistically different to wild-type controls (77.6 ± 5.7, *n* = 35, Kruskal-Wallis test adjusted for multiple comparisons) (Fig. 4B). There were also no statistical differences in the two other parameters we measured, egg output and hatching rate (Supplementary Fig. 3).

Additionally, the inheritance of the MDFS allele was extremely high, ranging from 99.5 ± 0.3% (*N* = 2569/2584, *n* = 34) in MDFS males to 99.7 ± 0.3% (*N* = 2416/2422, *n* = 35) in MDFS;X-shredder males while, as expected, the X-shredder allele followed Mendelian rules of segregation (Fig. 4C and D).

Next, we set out to measure the sex ratios in the progeny of the strains analysed. To this end, we reared the progeny from 14 wild-type females mated to MDFS males and 15 wild-type females mated to MDFS;X-shredder males and, subsequently, attributed the sex of the progeny at the pupal stage. The pupae from the wild-type control cross were easily identified as 57.0% male and 43.0% female (*N* = 1062) (Fig. 4E). In the experimental progeny, the sex ratio could not be inferred at this developmental stage, as all the pupae were presenting male-like features (*n* = 1041/1041 for MDFS and *n* = 1119/1119 for the MDFS; X-shredder), reflecting the dominant female sterility caused by the disruption of the *dsxF* allele. As such, the

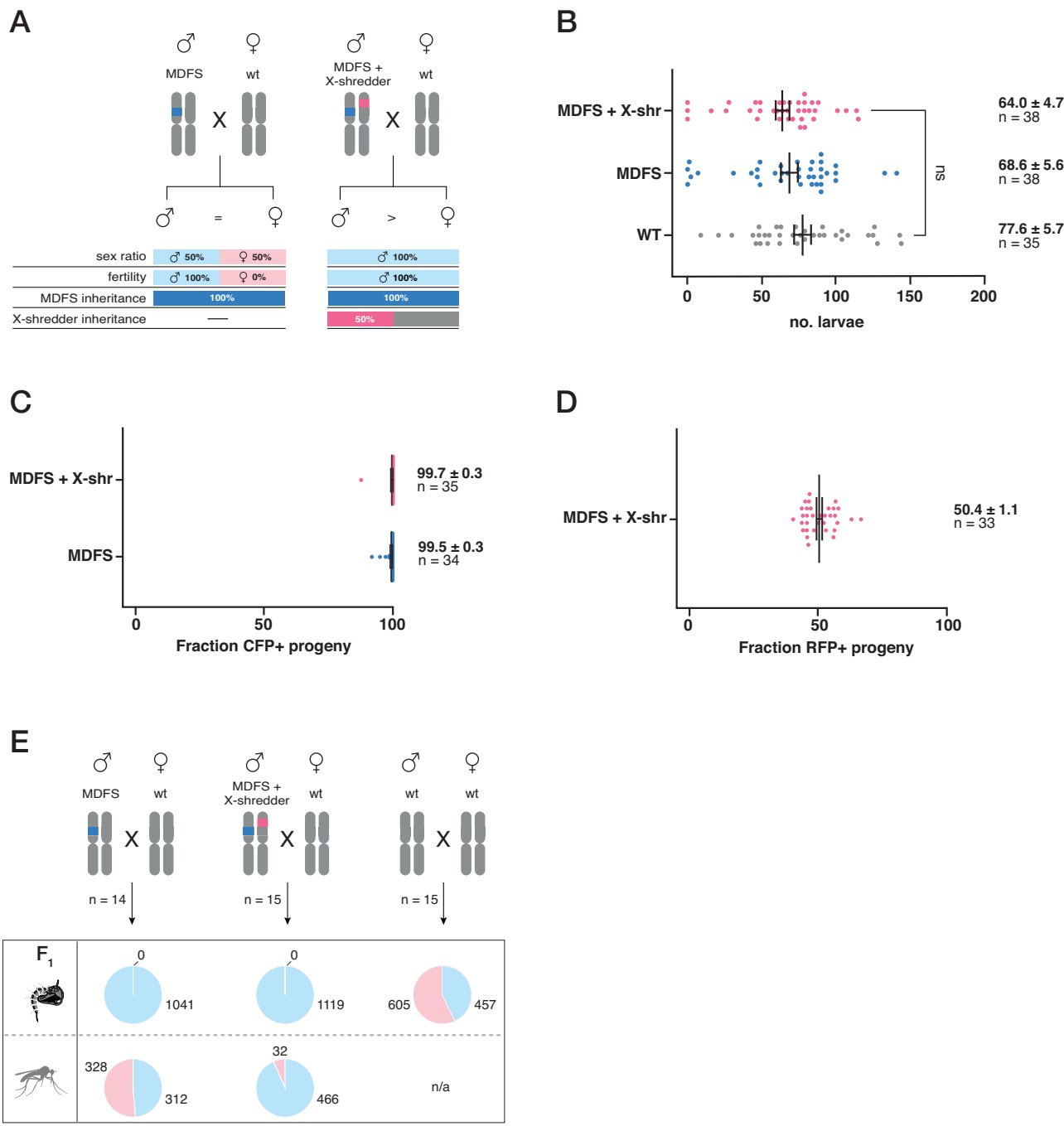

**Fig. 4 | Comparisons between MDFS and the MDFS + X-shredder males. A, left** In an ideal scenario, the MDFS construct is transmitted at super-Mendelian rates, and female progeny are sterile. **A, right** In an ideal scenario, the MDFS + X-shredder construct causes a strong male bias in the progeny (Galizi et al.). The MDFS transgene is marked with dark blue; the X-shredder transgene is marked with magenta. Figure was adapted from: iStock.com/LCOSMO. **B** Graph showing the fertility assay (larval output) of MDFS and MDFS + X-shredder males compared to wild-type males. MDFS and MDFS + X-shredder males were crossed to wild-type females that were allowed to lay singularly after a blood meal. No significant differences (ns) were found among the genotypes analysed (P > 0.05; Kruskal-Wallis test adjusted for multiple comparisons). **C** Graph summarising the inheritance rates of the MDFS transgene in the progeny derived from MDFS or MDFS + X-shredder males crossed to wild-type females. **D** Mendelian inheritance of the X-shredder allele in the progeny of MDFS + X-shredder males crossed to wild-type females. Black bars indicate mean and s.e.m. **E** Assessment of sex ratios in the progeny of MDFS and MDFS + X-shredder males. For each cross, 14–15 egg lays were selected. All the experimental progeny were male-like at the pupal stage, while the wild-type control had pupae showing both male and female phenotypes (first row of pie charts). At adulthood, the sex ratios were measured again in a subset of individuals, and a strong bias towards males was only found in the progeny of MDFS + X-shredder males (second row of pie charts). The male and female sexes are marked with cyan and pink colours, respectively. MDFS and X-shredder transgenes are marked blue and magenta, respectively. Vertical bars indicate the mean and the s.e.m. Source data are provided as a Source Data file. Figure was adapted from: iStock.com/LCOSMO.

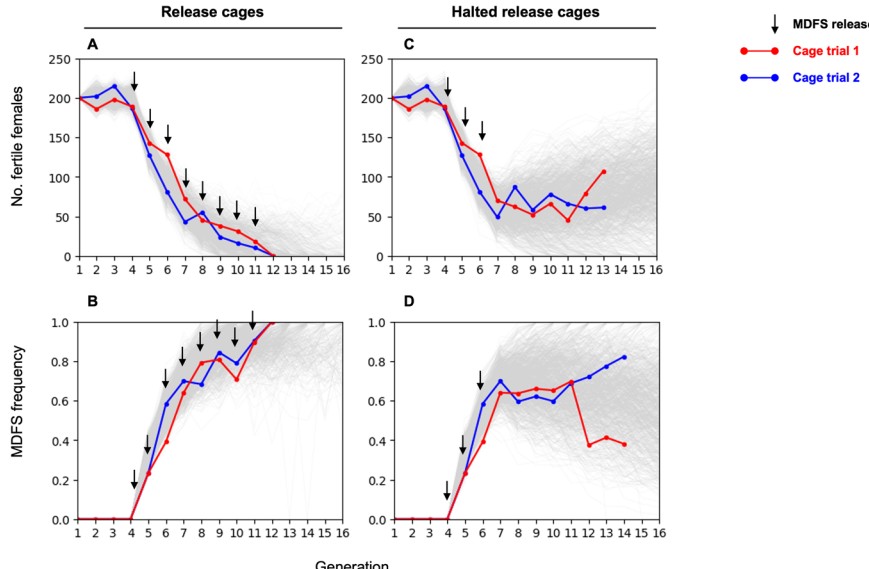

**Fig. 5 | Dynamics of the MDFS allele spread and effect on caged mosquito populations' reproductive capacity.** Two caged populations of 400 individuals each were maintained for three generations. At generation 4, 100 MDFS males were released in each cage every subsequent generation. **A** The number of fertile females dropped following each MDFS release and reached zero after the eighth release in both cages. **B** In parallel, the frequency of the CFP marker associated with the MDFS allele rose to 100%. **C** The number of fertile females and (**D**) the frequency of the MDFS allele followed a different trajectory when only three releases of MDFS males were conducted. In this case, neither of the two cages crashed, and the results fell within the expectations of the stochastic model. The red and blue lines depict replicates of the cage trial experiments, while the grey lines in the background represent the stochastic model simulations. Each black arrow represents one release of 100 MDFS males. Source data are provided as a Source Data file.

pupae from the experimental progeny were allowed to emerge, and a subset of adults was used to count the number of males and females (distinguishable at this stage) in each group. This analysis revealed sex ratios in line with the expectations, i.e., an approximately equal proportion of males and (intersex) females in the progeny of MDFS males (48.8% males and 51.8% females, $N = 640$) and a substantial male bias in the progeny of MDFS;X-shredder males (93.6% males and 6.4% females, $N = 498$) (Fig. 4E).

### MDFS exerts a self-limiting population suppression effect on caged mosquito populations

To evaluate the potential of the MDFS strain to suppress a target mosquito population, we employed a stochastic cage trial simulation model. This model incorporated the observed sterility of females carrying the MDFS construct, and the high rates of biased inheritance observed in the progeny of male mosquitoes. Our findings indicated that repeated releases of male hemizygotes at a ratio of 1:2 to the initial male population were expected to eliminate a caged population of 400 individuals (with equal numbers of males and females) after nine releases (mean = 9.22, standard deviation = 5.83, no. simulations = 1 million). Conversely, in the scenario of only three releases of male hemizygotes at this same 1:2 ratio, the population was expected to persist for at least 10 generations after the releases were halted, with a probability of 83% (no. simulations = 1 million). To mimic natural settings, in both cases, the mathematical model incorporated a population growth rate of ~9 adult females produced for every adult female[33].

A cage trial was performed in two replicate cages, initially seeded with an equal number (n = 200) of wild-type males and females. We maintained these two wild-type populations for the first three generations by randomly selecting 400 pupae to set up each new cage, and, in the process, we recorded the number of males and females. This first phase experimentally established our target mosquito population and recorded data before the release of MDFS males. At generation four, we initiated releases of 100 MDFS males in addition to the 400 wild-type pupae, corresponding to a 1:2 ratio to the initial wild-type male population or an allelic frequency of 10%. Following the third

release, each replicate cage was duplicated (four in total). Two cages were subjected to further generational MDFS releases (i.e., "release cage") while the two sister cages were maintained alongside it with no further releases (i.e., "halted release cage"). Throughout the experiment, at every generation, we recorded the frequency of the MDFS allele in the population (indicating the construct's spread) and the number of wild-type females in each cage (indicating its reproductive capacity).

The results of the cage trial were largely in line with expectations. In both release cages, populations collapsed after 8 consecutive releases of MDFS males. At that point, 100% of the cage trial populations carried the MDFS allele, and all females were intersex and thus unable to reproduce (Fig. 5A-B). Conversely, the populations in the halted-release cages did not collapse. They maintained a stable trend until generation 12, with differences between the two replicates but remaining within the stochastic model's expectations (Fig. 5C-D).

Interestingly, during the execution of the cage trial, we detected some rare females that did not inherit the MDFS allele (i.e., did not express the eCFP marker) but displayed somatic mosaicism. Because *vasa2* does not typically lead to Cas9 deposition from males[22,60], and given that sex-specific dominant negative mutations in the *doublesex* gene have been previously shown in *Drosophila*[58,68] and *An. gambiae*[45], we hypothesised this mosaicism was rather due to a dominant negative (DN) mutation at the *doublesex* locus generated as a result of an end-joining (EJ) event in the germline of the fathers. To investigate this hypothesis, eight eCFP-negative mosaic females were isolated from the progeny of the mosquito population in one of the release cages (generation 10), and Sanger sequencing was performed at the target site. All females carried an 11-bp deletion recently characterised as a DN mutation in the *doublesex* locus of *An. gambiae*[45], balanced with a wild-type allele (Supplementary Fig. 4).

### A population genetics model predicts that MDFS outperforms other self-limiting strategies

To assess the potential of the MDFS strain as a vector control tool for wild mosquito populations, we built a deterministic model and

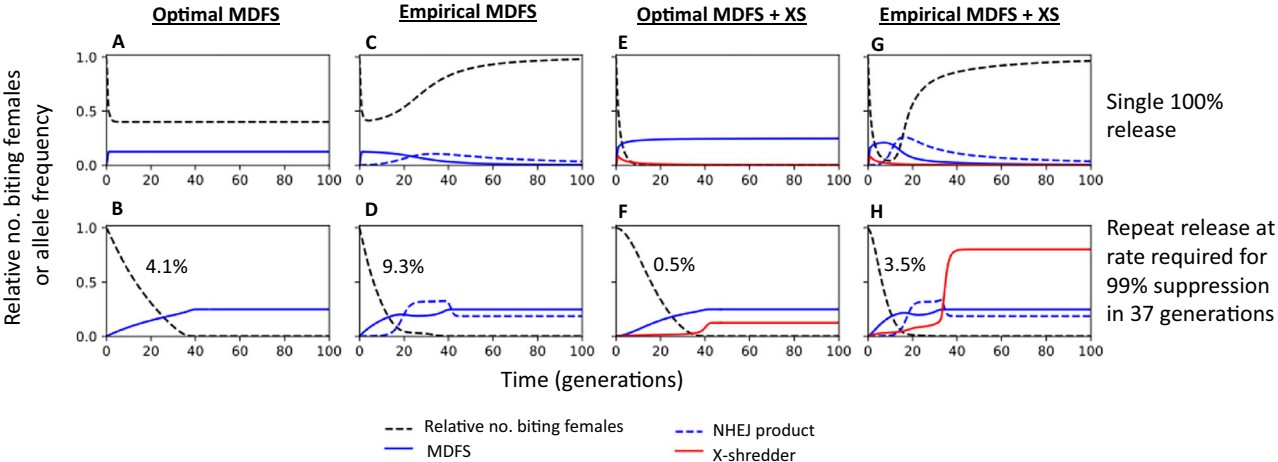

**Fig. 6 | Deterministic time series simulations of a population following the release of males carrying a single copy of an MDFS construct with idealised parameters. A, B** or parameterised using the empirical data (**C, D**) or an MDFS construct released alongside an X-shredder with idealised parameters (**E, F**) or parameterised using the empirical data (**G, H**). The top row shows a single 100% release of males relative to the starting population. The bottom row shows repeated releases each generation at the release rate required to suppress the population by 99% within 36 generations (release rates indicated in the figure). Each panel shows the relative number of biting females (black dashed) and the allele frequency in adults of the MDFS (blue, solid), the NHEJ product (blue, dashed, assumed to be recessive) and the X-shredder (red).

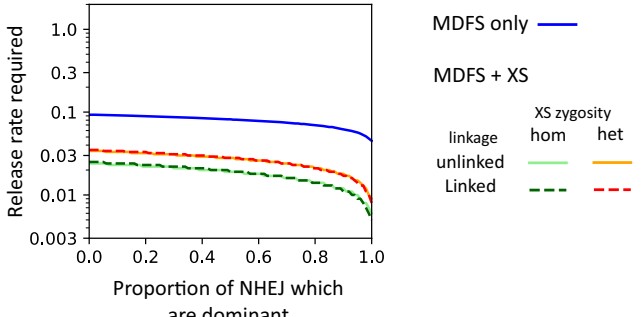

**Fig. 7 | Release rates required to suppress the population by 99% within 36 generations when releasing every generation as a function of the proportion of dominant NHEJ products.** Modelled constructs include the MDFS construct alone released in hemizygote males (blue) and the MDFS construct released with the X-shredder where males carry either a single copy of both constructs on different chromosomes (red) or a single MDFS and two copies of the X-shredder (green), whether the constructs are unlinked (solid, $r = 0.5$) or linked (dashed, $r = 0.01$). All baseline parameters are fixed at rates observed for the MDFS and XS constructs in the lab.

simulated its effect on the number of biting females in the target population. The simulations included scenarios with an idealised and empirical MDFS (data based on the traits observed in laboratory conditions, see Supplementary Methods) and incorporated the MDFS in combination with an autosomal X-shredder. Due to the difficulty in estimating the exact proportion of germline NHEJ products with recessive or dominant fitness effects, we initially adopted a conservative approach by assuming that the empirical MDFS construct yielded only recessive alleles.

A single 100% release of the optimal strategies would cause a level of population suppression that would remain stable over time, as would the frequencies of the transgenes (Fig. 6A and E). On the other hand, in the empirical scenario, where homing and male bias have values lower than the optimal 100%, the female population would rebound to pre-release levels, and the frequency of the transgenes would reduce over time (Fig. 6C andG). This is largely due to the appearance of recessive NHEJ products, which confer a

fitness advantage when compared to the dominant nature of the MDFS allele.

With repeated releases, 99% population suppression can be achieved and maintained in both the optimal and empirical scenarios. However, as with the single releases, recessive NHEJ products appear in the latter but not in the former (Fig. 6B, D, F and H). Interestingly, a proportion of recessive NHEJ products reaches a stable equilibrium in both empirical models despite the high level of population suppression. If some of the NHEJ products created by the MDFS construct have dominant fitness effects (as observed during our cage trial), the release rate required (i.e., number of males needed) to achieve this level of suppression can be reduced (Fig. 7, blue). Though the hemizygous MDFS females we observed were fully sterile, it is possible that some residual fertility in hemizygous females would cause the construct to drive and therefore no longer be localised. Using a simple deterministic model developed by Deredec et al., we found that if all homozygous females are fully sterile and males are fully fit, drive can be prevented if $h > 2 - 1/d$, where $h$ is the fitness cost in hemizygous females and $d$ is the transmission rate from hemizygous males and females (assumed to be equal; d = ½ corresponds to be Mendelian inheritance)[36]. In our MDFS strain $d = 0.9955$ and drive would not occur if hemizygous females displayed a fitness cost, $h$, greater than 0.9955.

Modelling data suggest that an MDFS strategy and its combination with the autosomal X-shredder are markedly more efficient than other self-limiting strategies, even when the former are modelled using empirical parameters and the latter using optimal ones.

Specifically, for a 99% population suppression within 36 generations, the MDFS strategy would require 13-fold lower releases than an optimal SIT strategy, 4.7- and 7.3-fold lower releases than optimal RIDL and fs-RIDL technologies, respectively, and 4.3-fold lower releases than an X-shredder (Table 1). In the scenario where the MDFS males also carry an autosomal X-shredder construct, release rates can be reduced to 0.035 ( > 60% reduction compared to 0.093 when using the MDFS alone) or to 0.025 if released with two copies of the X-shredder ( > 70% reduction compared to the MDFS alone), regardless of the extent of linkage between the two constructs (Table 1, Fig. 7).

## Discussion
Here, we describe a self-limiting genetic strategy for mosquito population suppression. Specifically, we have developed a strain of *An.*

**Table 1 | The number of released males as a proportion of the initial male population required to suppress the population by 95 or 99% within 36 generations**

| Level of pop. suppression (%) | Optimal SIT | Optimal RIDL | OptimalfsRIDL | Optimal X-shredder | Optimal MDFS | Optimal MDFS + XS | Empirical MDFS | Empirical MDFS + XS |
|---|---|---|---|---|---|---|---|---|
| 95 | 1.297 [14.6] | 0.438 [4.9] | 0.668 [7.5] | 0.396 [4.5] | 0.037 [0.42] | 0.004 [0.05] | 0.089 | 0.025 [0.28] |
| 99 | 1.3 [13.0] | 0.440 [4.7] | 0.678 [7.3] | 0.403 [4.3] | 0.041 [0.44] | 0.005 [0.05] | 0.093 | 0.035 [0.38] |

[Strategy/Empirical MDFS].
The table also shows the release rates for alternative optimal strategies, including males carrying two copies of a dominant lethal gene which causes death in both males and females before (SIT) or after (RIDL) density-dependent mortality or only females after density-dependent mortality (fsRIDL) and releasing two copies of a sex-ratio distorter which causes males to produce all Y-bearing sperm (X-shredder). Values in square brackets show the proportional increase in release rate requirements when comparing each strategy to the empirical MDFS alone.
Release rates for males carrying a single copy of the MDFS construct with or without a single X-shredder construct are shown when parameterised based on the optimal or empirical case.

*gambiae* called MDFS, which carries a genetic construct that is transmitted to offspring at nearly 100% frequency. The MDFS construct has a dominant effect that leads to the sterility of the female progeny.

The high rate of inheritance is consistent with previous research on homing gene drives in this same species that also utilised the regulatory sequences of *vasa2* [22,50,51,62].

The female sterility feature is achieved through Cas9-induced mutagenesis at the female isoform of the *doublesex* gene (*dsxF*). *Doublesex* is a key regulator of the sex determination pathway, and it is responsible for sexual dimorphism, particularly marked in mosquito species [34,35]. We have previously shown that knockout of *dsxF* in *An. gambiae* results in genetic females exhibiting an intersex phenotype and being fully sterile [23]. In this study, mutagenesis of *dsxF* is due to the leaky expression of Cas9 nuclease driven by the *vasa2* regulatory sequences, leading to partial conversion of the soma to homozygosity for a null allele.

While hemizygous MDFS females are completely sterile, there are some observable differences in their phenotype compared to *dsxF* knockout females. Specifically, MDFS females have rudimentary ovaries, which are absent in *dsxF* knockout females. We believe this difference arises because, in *dsxF* knockout females, the gene's function is disrupted in the entire organism. In contrast, mutagenesis at the *dsxF* locus in MDFS females occurs only in part of the soma and throughout female development (i.e., females are born hemizygous for the null allele), leading to an intersex phenotype that is less penetrant.

This hypothesis is supported by sequencing data of MDFS individuals at the *doublesex* locus, which shows a significant decline in the wild-type allele starting from the larval stage (Supplementary Fig. 2). This suggests that the leaky expression of the *vasa2* promoter in the soma may start early in development, consistently with the early expression profile of the endogenous gene *vasa* [69-72].

In the cage trial, the MDFS allele reached a frequency of 100%, and no fertile females were left after eight MDFS releases at a ratio of 1:2 compared to wild-type males. This suggests that no resistant mutations restoring the function of *doublesex* gene emerged. If such mutations had occurred, fertile females would have been selected for, leading to a decrease in the frequency of the MDFS allele over time. Such results confirm that the intron4-exon5 boundary of the *doublesex* gene is highly functionally constrained and, as such, constitutes an optimal target for genetic population suppression strategies. Nonetheless, if MDFS were chosen for further development and considered for deployment in the field, it would be advisable to include at least one additional gRNA that targets the same *doublesex* locus in a nearby sequence. This strategy was recently used to develop a next-generation gene drive targeting highly conserved and non-overlapping sites in the female-specific exon of the *doublesex* gene in *An. gambiae* and *An. stephensi* [73,74]. Such a system was shown to actively eliminate resistance alleles, and mathematical modelling suggests that this gene drive could reduce resistance across large populations of wild malaria mosquitoes, effectively suppressing them [73]. Multiplexing of gRNAs is becoming increasingly common in most homing-based approaches [27,55,56,73,74]. However, the number of gRNAs to include would

need to be balanced against the construct's overall efficiency and stability, as the loss of one or more gRNAs during the homing process has been previously observed [27,73,75,76]. Possible solutions have been proposed, including interspacing gRNAs with tRNA scaffolding [27,56,77] and flanking gRNAs with self-cleaving ribozymes [78].

Both release cages showed a similar dynamic for the MDFS construct spread, and their populations crashed at the same generation. The strong correlation between the cage trial results and the model's predictions indicates a precise capture of the parameters influencing its dynamics, at least when released in small cages.

In contrast, the populations in cages where releases of MDFS males were stopped persisted stably for over 8 generations without collapsing, which aligns with the modelling predictions.

The cage trial allowed us to compare the in-vivo results of MDFS with those of other self-limiting strategies. We have previously shown that an autosomal X-shredder system can eliminate caged mosquito populations within six generations when released at an over-flooding ratio of 3:1 [14]. This release ratio is six times larger than the 1:2 ratio we employed in the MDFS cage trials (100 MDFS males for 200 wild-type males). Interestingly, this comparison aligns closely with findings from our deterministic model, which predicts that the empirical MDFS is 4.3 times more efficient than an optimal X-shredder. This suggests that the results presented here are robust.

Notably, mathematical modelling indicates that using an X-shredder system with MDFS could lower release rates to less than 40% of those required for MDFS alone.

In the future, estimating the release rates needed for field trials will require more detailed models that consider several factors, including additional parameters that were not measured in this study (e.g., male mating competitiveness, survival, etc.), the effects of migration and spatial dynamics. Migration can limit the spread of genetic drives in the target population by consistently introducing wild-type individuals, and it may also lead to the transgene's introduction into neighbouring populations [79].

Additionally, future models should consider the various repair outcomes at the cut site (wild-type allele, partial homing, or recessive or dominant NHEJ products), which we did not include due to their rarity.

Crucially, the dominant effect of MDFS on females is essential for the self-limiting of this strategy; if some MDFS females were not fully sterile, they could potentially contribute to the spread of the MDFS construct, converting MDFS into a self-sustaining strategy (i.e., a recessive CRISPR-homing gene drive). An improved version of MDFS could be developed by targeting a highly conserved haplo-insufficient female fertility gene instead of a haplo-sufficient one (providing its phenotype is strictly restricted to females). This approach would eliminate the selection of recessive NHEJ alleles at the expense of the MDFS transgene, as any mutations would be dominant and, therefore, would no longer have a selective advantage over the MDFS allele. Additionally, targeting a haplo-insufficient gene could enhance the self-limiting nature of MDFS, as the genetic construct in hemizygous females would already provide a dominant fitness cost.

A potential female-specific haplo-insufficient gene has been identified in *Aedes aegypti*[80]; however, none has been found in *An. gambiae*. Investigating this area could provide valuable insights for future research into self-limiting strategies aimed at population suppression in this species.

In addition, even though no MDFS fertile females were detected in this study or during routine mosquito husbandry to maintain the strain in our facilities, future research involving larger populations could provide further insights into the potential fertility of any MDFS females. Moreover, the possibility of MDFS evolving into a self-sustaining gene drive system would need to be considered as part of an environmental risk assessment system to help inform the decisions regarding potential field trials for this technology[81].

Finally, to effectively bridge the laboratory and field testing of MDFS males, either alone or combined with the X-shredder, strategies for large-scale rearing and sex sorting[82–84] will need to be developed. In the current design, large-scale production of males carrying the MDFS or X-shredder alleles is limited by the inability to maintain these strains in homozygosity, as females carrying these constructs are either sterile (MDFS) or rare (X-shredder). Additionally, the morphological similarity of male and female MDFS pupae complicates efficient sex sorting at scale. Potential solutions include the use of sex-linked genetic markers for early sex discrimination, implementation of automated fluorescent sorting technologies (e.g., COPAS) for high-throughput isolation of MDFS or MDFS;X-shredder individuals, and controlling effector expression through inducible or repressible genetic systems.

## Methods

### Generation of the MDFS allele

The MDFS allele was generated in vivo by φC31 recombinase-mediated cassette exchange (RMCE) using construct pAS501, which encompassed the *hCas9* and the *dsx* gRNA transcription units, as well as reporter actin5c::eCFP cassette within two reversible φC31 *attB* recombination sequences. To make pAS501, plasmid p165[22] was digested with restriction enzymes to isolate a fragment comprising the backbone, the *hCas9* flanked by the *vasa2* regulatory regions and a U6::gRNA cassette containing a spacer cloning site. Next, the actin5-c::eCFP::SV40T marker cassette was amplified from the 32701 plasmid[85] and the two fragments ligated by Gibson assembly. Finally, we inserted the gRNA targeting the previously described target site at the *doublesex* intron 4-exon 5 junction using Golden Gate cloning, as detailed by Kyrou et al, 2018.

### Embryo microinjection and selection of transformed mosquitoes

All mosquitoes were reared under standard conditions of 80% relative humidity and 28 °C. Females were blood-fed with bovine blood (with added heparin) using a Hemotek membrane feeding system, and freshly laid embryos were aligned and used for microinjections as described previously[86]. To generate the MDFS allele, embryos from the *dsxF⁻* knock-in line containing an eGFP φC31 acceptor construct at the *dsx* target site[23] were injected with a solution containing pAS501 (at 50 ng/µL), the C77 plasmid that expressed the AcrIIA4 anti-Cas protein (this was used at 10 ng/µL with the intent to reduce any potential Cas9-induced toxicity)[87] and a plasmid-based source of φC31 integrase (at 200 ng/µL)[88]. All the surviving G$_0$ larvae were crossed to wild-type mosquitoes and G$_1$ positive transformants were identified using a fluorescence microscope (Eclipse TE200) as CFP+ larvae.

### Maintenance and containment of mosquitoes

All mosquitoes were housed at Imperial College London in an insectary that is compliant with Arthropod Containment Guidelines Level 2 (ACL2). All GM work was performed under institutionally approved biosafety and GM protocols. Moreover, because of its location in a city with a northern temperate climate, *An. gambiae* mosquitoes housed in the insectary are also ecologically contained. The physical and ecological containment of the insectary is compliant with guidelines set out in a recent commentary calling for safeguards in the study of synthetic gene drive technologies.

### Molecular confirmation and validation of MDFS cassette integration

Successful RCME of the MDFS cassette into *Agdsx* at exon 5 was confirmed by PCR, using genomic DNA extracted from three adults using the Wizard Genomic DNA purification kit (Promega), with one primer binding the eCFP in the MDFS cassette (CACTACCTGAGCACCCAGTC) and the other binding the neighbouring genomic integration site (ACATTGTCGTCTCAACTCCCA).

### Phenotypic characterisation and microdissections

Microdissection and phenotypic characterisation were carried out using an Olympus MVX10 optical microscope. Mosquitoes were collected in Eppendorf tubes and anaesthetised on ice for 5 min before being dissected in PBS solution (1X, pH 7.4). Images of mosquito heads, cerci, and claspers were taken with a Mitutoyo 5X Plan Apo lens mounted on a Nikon bellow connected to a Nikon D800 camera. The images were stacked using ZereneStacker v1 software. Stacks were taken every 10–20 µm. Adobe software (Lightroom Classic and Photoshop 2024) was used to adjust brightness, contrast, and crop images.

### Pooled amplicon sequencing at *doublesex*

Pooled amplicon sequencing at the *doublesex* T1 site was performed on pooled L1 larvae ($n = 2$), individual male and female L4 larvae ($n = 6$ per sex), individual male and female pupae ($n = 6$ males and $n = 5$ females) and individual male and female adults ($n = 6$ per sex). Each L1 sample consisted in the entire progeny (>100 individuals) of a replicate cross of >30 MDFS males and >30 wildtype females. The L1s were not screened to remove the rare CFP- individuals, since these represented ≈0.4% of the progeny. Genomic DNA was extracted using the Wizard® Genomic DNA Purification Kit (Promega) (L1 samples) or with the DNeasy Blood & Tissue Kit (QIAGEN) (other samples). The PCR and analysis of the raw sequencing data were conducted essentially as described in Kyrou et al. Briefly, the genomic DNA was subjected to PCRs that were performed under non-saturating conditions and that amplified a 286 bp fragment spanning the *doublesex* T1 target site, using the primers dsx_poolamp_F. (ACACTCTTTCCCT ACACGACGCTCTTCCGATCTACTTATCGGCATCAGTTGCG) and dsx _poolamp_R (GACTGGAGTTCAGACGTGTGCTCTTCCGATCTGTGAAT TCCGTCAGCCAGCA) containing partial Illumina adaptors (underlined). The reactions were then PCR purified using the Monarch® PCR & DNA Cleanup Kit (NEB). Next, the DNA concentration of each sample was quantified using a NanoDrop spectrophotometer and subsequently normalised to 20 ng/µL, before being submitted for the Amplicon-EZ sequencing service (Genewiz). The resulting amplicon sequencing reads were then analysed using the CRISPResso version 2.0.29.

### Fertility assay of MDFS and MDFS;X-shredder males

The fertility assay was carried out essentially as described before[23]. Briefly, groups of 50 male mosquitoes from each of the three genotypes (MDFS, MDFS;X-shredder and wild-type controls) were mated to an equal number of wild-type females for 6 d, blood-fed, and a minimum of 42 females for each group were allowed to lay individually. The entire egg and larval progeny were counted for each lay and the entire larval progeny were screened for presence of eCFP. Females that failed to give progeny and had no evidence of sperm in their spermathecae were excluded from the analysis. Statistical differences between genotypes were assessed using the Kruskal–Wallis test adjusted for multiple comparisons.

### Fertility assay of MDFS females and sex ratio verification

Part of the progeny resulting from the fertility assay described above was maintained to adulthood to measure the relative fecundity of female mosquitoes carrying the MDFS allele and the presence of male bias. Specifically, a minimum of 14 lays were kept for each cross, and each lay was reared in a separate tray. At adulthood, 50 females were taken at random from the lays of the wild-type males and, separately, of the MDFS males, and 15 were taken from the lays of the MDFS;X-shredder males (all those that could be retrieved), and each group was crossed to 50 wild-type males. In parallel, the remaining progeny of the MDFS lays ( > 1200 individuals) were allowed to emerge in a cage and mated to each other. After 5 d, the four cages were blood-fed and females were allowed to lay on a petri dish lined with filter paper. The eggs were photographed and counted with the EggCounter software. Finally, the entire lays of the wild-type controls were sexed at the pupal stage, while the lays of the MDFS and MDFS;X-shredder males were frozen at adulthood and later sexed to determine their sex ratio.

### Cage trials

Two cage trials were initiated using 400 wild-type pupae, which, for the first generation only, were sexed as 200 females and 200 males. The cages were maintained for three generations by randomly selecting 400 pupae from the progeny of each new generation. From the fourth generation onwards, 100 MDFS males were released as pupae in each cage, alongside the 400 unsexed pupae of the cage trial population. From the fifth generation onwards, 600 L1 larvae were selected at random and screened for the eCFP marker to determine the frequency of the MDFS allele, and the 400 pupae used to seed the next generation were sexed to determine the number of morphological looking female as a proxy for fertile females. In the generation following the third MDFS release, the two cage trial populations were duplicated, for a total of four cages. In two of these (one per experimental group) the MDFS releases continued as described above, until there were no fertile females left (i.e., no progeny produced), while the other two cages were maintained in parallel, as described above, except without further MDFS releases.

The cage trials were maintained using the following cycles: pupae were allowed to emerge in the cage and were blood-fed one week later with bovine blood (with added heparin) using Hemotek feeders. On the third day PBM, females were allowed to lay in in Pyrex 100 mL glass containers filled with ≈90 mL DM water with salt, with a folded filter paper placed on top for the egg collection. The next day, the eggs were gently sprayed into plastic trays filled with -500 mL DM water with salt and lined with filter paper. Two days later, 600 newly hatched L1 larvae were screened and split into trays of 150 larvae each. The larvae were then reared for 7-9 days to the pupal stage, at which point 400 were randomly selected to seed the new cages.

Since MDFS female pupae are indistinguishable from males, the MDFS line was maintained in parallel to the cage trial populations by crossing it every generation to a line homozygous for a *β2*::mCherry marker. This allowed the sexing of MDFS males to be released from their female siblings, as they were easily recognisable through their testes expressing mCherry. The three maintenance cages needed to achieve this (the marker cage, the MDFS cage, and the cage were MDFS and marker individuals were crossed) were maintained in parallel and under the same conditions as the cage trial cages.

A mortality rate was applied each generation to approximate a natural population growth rate of nine females produced per female[33] by keeping a quarter of the L1 larvae and discarding three-quarters each generation. The rationale for this approach was as follows. Based on the fertility assay conducted, each female mated to an MDFS male lays an average of 92.64 eggs with a hatching rate of 86.2%. Also, based on an analysis of the cage trial trays, the probability of survival of the hatched larvae to adulthood is 87.98% in cage trial conditions. Finally, the probability of mating of the females that reach adulthood is 93.3%.

These values result in an Rm (i.e., reproductive rate) of 32.77 females produced per adult female, which, if divided by four, results in 8.19 females produced per female, approximating the Rm used by Galizi et al. when assessing a sex-ratio distortion system[14].

### PCR of females harbouring dominant negative mutations

8 females that did not express CFP but displayed somatic mosaicism were isolated from the backup trays of the cage trial 2 (CT2) release cage, generation 10 (G10). They were frozen at adulthood, and the *dsx* target site was amplified by PCR with primers CACTCTTTCCCTA-CACGACGCTCTTCCGATCTACTTATCGGCATCAGTTGCG and GACTGGAGTTCAGACGTGTGCTCTTCCGATCTGTGAATTCCGTCAGCCAGC A using the Phusion High-Fidelity PCR Master Mix with HF Buffer (NEB) and sent for Sanger sequencing.

### Mathematical modelling

We developed two types of mathematical models: a) a stochastic model to simulate small cage populations, guide trial design, and predict outcomes and b) a deterministic model to explore the impact of releasing the genetic constructs into a wild population. The models are described in detail in Supplementary Methods, and all code is available on GitHub via (https://github.com/KatieWillis/MDFS).

### Statistics and reproducibility

The statistical tests and sample sizes used in each experiment are described in the corresponding methods section. No statistical method was used to predetermine sample size, which was chosen consistent with the previous literature reporting similar assays. Sample size was maximised within the feasibility of performing biological assays with live insects. No data were excluded from the analyses.

### Reporting summary

Further information on research design is available in the Nature Portfolio Reporting Summary linked to this article.

## Data availability

Amplicon sequencing data generated and analysed in this study are available in the Sequence Read Archive (SRA) with the accession code PRJNA1227481. Source data are provided with this paper.

## Code availability

Code used for the simulations performed in this study is fully available on GitHub: (https://github.com/KatieWillis/MDFS), under a GNU General Public License 3.0 (GPL-3.0).

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

## Acknowledgements

We would like to thank Andrew Hammond for his helpful suggestions on experimental design. We also thank John Connolly, Silke Fuchs and Alekos Simoni for their valuable feedback. The Gates Foundation, Open Philanthropy and the Sir Keith and Lady Rita O'Nions PhD scholarship supported this work.

## Author contributions

A.C. conceived the project; A.C., B.F., A.S. and F.B. designed the research; A.S., K.W., H.R.G. and M.G. performed research; A.S., K.W., B.F. and F.B. analysed data; K.W. and A.B. performed mathematical modelling; and A.S. and F.B. wrote the paper with input from all authors.

## Competing interests

A.C. is a founder of Biocentis, Ltd. A.S. is an employee at Biocentis. The remaining authors declare no competing interests.
