## [Transparent Peer Review File · Nature Communications]

A male-drive female-sterile system for the self-limited control of the malaria mosquito *Anopheles gambiae*

Corresponding Author: Dr Federica Bernardini

Version 0:

Reviewer comments:

Reviewer #1

(Remarks to the Author)

This work from the Crisanti group provides an additional self-limiting genetic control for the control of malaria mosquitoes. This's been cleverly designed by using preexisting target sites and nuclease expressing constructs, i.e. dsx from Kyrou 2018 and X-lined rDNA from Galizi 2014, to develop a dominant female sterile approach. The transgenes are passed on at supermendelian frequencies exclusively via males, whilst females develop an intersex phenotype due to leaky expression of the CRISPR nuclease targeting the female-specific isoform of dsx or their proportion is further reduced in the case the X-targeting nucleases is also expressed during male meiosis. The manuscript is well written and both experimental design and analysis/discussion look waterproof to me. I only have a few minor comments and suggestions noted below.

temporally and spatially restricted version of a homing gene drive 61 > The reference used doesn't seem the most appropriate here.

the germline of both sexes and exhibits somatic leakiness 21,50,51.
> There might be other relevant works to be referenced here, particularly those looking at gene drive resistance.

vasa2 promoter would cause the expression of Cas9 in somatic cells.
> Has this been clearly confirmed vs deposition or only assumed?

two attP docking sites at the doublesex intron4-exon5 boundary (Kyrou et al., 2018; Supplementary Figure 1 and Figure 2A).
> It might be useful to briefly explain the difference vs similar iterations of this, e.g. Kyrou 2018 and Hammond 2016

subsequent molecular investigation confirmed the successful recombinase-mediated cassette exchange with the target locus (Supplementary Table 1).
> I don't see the molecular confirmation in Sup Tab 1, might be wrong reference?

Additionally, male accessory glands (MAGs) were present in all females, though in a small subset, they seemed less developed (e.g. there was only one or both were lighter in colour) (n=8/41) (Figure 3J).
> Is this the same phenotype shown in Kyrou 2018?

that the penetrance of the dominant sterile phenotype, by virtue of the vasa2 promoter, has remained stable over time.
> Data shown where/not shown?

This scenario allows the MDFS allele to transiently drive in the population, as it would be passed on by a greater than 50% portion of the progeny (i.e., the males).
> might be more useful earlier on in this section, e.g. near ref 43?

were not statistically different to wild-type controls (77.6 ± 5.7 , n=35) ...and following fertility assays
> These differences are not significant due to low sample size - if that would be increased (e.g. as expected in field populations) and average N maintained the difference would be significant. This should be addressed in the text.

The simulations included scenarios with an idealised...

> Please provide additional details, e.g. I don't see EJ in the ideal model?

This is largely due to the appearance of recessive NHEJ products, which confer a fitness advantage when compared to the dominant nature of the MDFS allele.

> What about fitness, would that have an impact too?

However, as with the single releases, recessive NHEJ products appear in the latter but not in the former (Figure 6B, D, F and H).

> See my previous comment about EJ, it would not be surprising if simply not included in the optimal model?

regardless of the extent of linkage between the two constructs (Table 1, Figure 7).

> Rates in the table and %s used in the figure do not help to compare the two. Possibly they could even be merged as unique figure

(i.e., females are born hemizygous for the null allele), leading to an intersex phenotype that is less penetrant.

> what would be the impact of low or null penetrance hence fertile females? Would the transgene drive at that point?

The cage trial allowed us to compare the in-vivo results of MDFS with those of other self-limiting strategies.

> is that the case? I could not find other constructs tested in the cage trial, maybe this should refer to the models?

(Remarks on code availability)

Reviewer #2

(Remarks to the Author)

In this study, the authors make a dominant female-sterile homing drive in *Anopheles gambiae* for self-limiting population suppression. The drive works as expected, and mathematical modeling shows that this strategy is very promising for future field use.

Overall, I like the study, but there are several factors that reduce its impact:

1. The idea was proposed by the lab several years ago after an experimental demonstration, and these systems have also been modeled before.
2. A recent study (see below) made a similar drive in *A. stephensi*. It was a split drive, but had an important feature of two gRNAs.
3. This drive only has one gRNA. As indicated by a preprint that is referenced here, this means that the drive in this study can't be considered as a release candidate.

On the other hand, I appreciate the cage study, and the complete drive in a new species is also a useful advance. Thus, I support publication of the study in *Nature Communications* after some minor revisions.

Line 18: "Propose" may not be the best word to use in the context of this. As far as I know, the idea of male drive female sterile was originally proposed in Hammond 2016, and since then, two of our recent papers (Xu et al 2025 in *Nature Communications* and Han et al 2025 in *Molecular Biology & Evolution*) have elaborated on the idea quite a bit. Indeed, these two studies have several relevant features and should probably be incorporated throughout the manuscript (in particular, Xu et al made a split form of MSFD in *Anopheles* that had good performance).

Line 37: Should probably have a reference for this.

Line 53-55: This part could be made more concise.

Line 64: Not sure that "Contained" means in this sentence.

Line 77-80: It's not just this. The gene drive will simply be more efficient at overall suppression (including suppressing the number of fertile females) if it targets a female-specific gene rather than a both-sex haplosufficient gene.

Line 119: Reference 46 was a self-sustaining homing drive. Also, I'm not sure if the integral gene drive references here should necessarily be considered as self-limiting split drives, even though they do have a "split" aspect to them.

Line 128: "shredding" should probably be "shredder".

Line 152: These two papers indicate that HDR may occur at fairly high rates (perhaps even the dominant form of repair) in flies, so this statement may be not quite right. Of course, for this drive, it doesn't matter whether HDR or end-joining occurs. Female mosquitoes would still be expected to become sterile.

Li Z, Marcel N, Devkota S, Auradkar A, Hedrick SM, Gantz VM, Bier E. CopyCatchers are versatile active genetic elements that detect and quantify inter-homolog somatic gene conversion. *Nat Commun*, 12, 1–12, 2021.

Du J, Chen W, Jia X, Xu X, Yang E, Zhou R, Zhang Y, Metzloff M, Messer PW, Champer J. Germline Cas9 promoters with improved performance for homing gene drive. *Nat Commun*, 15, 1–14, 2024.

Note that this affects the interpretation of Supplementary Figure 2 (it just shows the fraction of wild-type for alleles that have not been converted to drive).

Line 204: Why not show it if you have this data? This is an important target site, so knowing exactly what resistance alleles look like could be interesting.

Line 217-224: A brief word about getting the X-shredder in females would be useful here, for readers wondering how it would be possible (presumably imperfect efficiency means that it sometimes finds itself in females, which remain fertile due to male-specific expression).

Line 291: Yadev et al. made a dominant-sterile drive targeting doublesex (like this manuscript), among other drives, but dominant-sterile resistance alleles were not identified.

Line 347: "and fully" should be "and being fully".

Line 367: "doublesex gene is highly functionally constrained" it would be more accurate to say the target site at dsx intron-exon boundary is constrained. Other target sites may not be as conserved and useful for gene drives.

Line 374: Xu et al. (2025, Nature Communications) also showed this in *A. stephensi*.

Line 378: Another counterbalancing feature is that drive efficiency will likely go down with more than two gRNAs, though how much is unclear (S. Champer et al., Science Advances 2020).

Line 420: References 57-58 (and your dominant-sterile resistance allele) may offer a solution - expressing the male isoform of dsx in females.

Line 430: "In vivo" should be italicized.

Line 436-437: It's not good to reference an unpublished plasmid. If you used it in the manuscript, make the sequence available. At minimum, all the relevant annotated plasmid sequences in this study should be posted.

Line 553-541: Can the numbers this was set to be justified in any way? It's okay if there is no firm reference, just saying something approximate would be sufficient.

Line 884: "Trail" should be "trial".

References: Species names are not italic. Reference 57 has been published in Nature Communications.

Figure 2: The text in the image is a bit small.

Figure 3: The labels E, F, G and I, H, J are reversed in the legend. Is the magnification the same for I, H, and J? Maybe make this clear in the legend.

General considerations for Figure 6-7 and Table 2: These may be somewhat optimistic when considering practical combination of X-shredder and MDFS. It is often difficult to get an X-shredder in females. This limits your options for consistently generating homozygous individuals for release. In fact, a perfect X-shredder would make this even more difficult, requiring direct injection rather than lucky crosses. In this case and probably even for practical purposes for an imperfect X-shredder, you would probably still mostly be crossing males with a single drive and X-shredder copy to wild-type females in order to generate your release batch. Only half the male progeny will carry the X-shredder. You can keep using a few of these to stably generate the next generation, but unless you want to discard half your males, your released males would still only have an X-shredder half the time. To avoid this, you'd need some sort of induction system. This could be mentioned, and it might also be useful to model releases where only half the males have X-shredders. It will still be helpful compared to just MDFS alone.

In this study, I don't remember seeing anything about male mating success fitness. This wasn't measured. It's probably not a huge issue, but still worth mention that some additional fitness costs may somewhat slow down the drive.

Sincerely,
Jackson Champer

(Remarks on code availability)
Code is fine.

Reviewer #3

(Remarks to the Author)

(Remarks on code availability)

Version 1:

Reviewer comments:

Reviewer #1

(Remarks to the Author)

The authors have addressed my concerns and I don't have any other comments other than the two minor suggestions below - being only minor suggestions I don't need to see the revised manuscript again.

C1 vasa2 promoter would cause the expression of Cas9 in somatic cells.
> Has this been clearly confirmed vs deposition or only assumed?

R1 We have previously demonstrated that the vasa2 promoter results in high levels of end-joining repair in the embryo following maternal deposition of the endonuclease. Additionally, we reported that the expression of the endonuclease driven by this promoter outside of the mosquito germline—specifically, leaky expression in somatic tissues—caused significant unintended fitness costs in females carrying a single copy of the gene drive targeting a female-specific gene. This cost arose from the partial conversion of somatic cells to homozygosity for a null allele (Hammond et al., 2017 and 2021). In the context of the MDFS strategy, the transgene is always passed on to the offspring through males, and the deposition of endonucleases driven by the vasa2 promoter has not been observed paternally. The dominant effect of the MDFS in the female offspring is attributed to the leaky expression of nucleases in the somatic tissues.

C2 “The deposition of endonucleases driven by the vasa2 promoter has not been observed paternally” > It looks like this was not measured here and therefore the assumption may be derived from elsewhere? I would also assume that deposition would likely depend on dose of germline expression and nuclease thermostability. I still don't see how paternal deposition could be excluded here - unless provable otherwise I would recommend adding the paternal deposition as possible source/cause of somatic activity.

C1 subsequent molecular investigation confirmed the successful recombinase-mediated cassette exchange with the target locus (Supplementary Table 1).

> I don't see the molecular confirmation in Sup Tab 1, might be wrong reference?

R1 We appreciate the reviewer's feedback. The incorrect reference to Supplementary Table 1 has been removed, and we have now included the PCR results confirming the integration of the transgene at the doublesex locus in Supplementary Figure 1.

C2 Might be useful to show the primers used in the schematic above to make the gel photo meaningful.

(Remarks on code availability)

Reviewer #2

(Remarks to the Author)

The revisions are mostly quite good, and I think that the manuscript can be accepted now. We have a few more small comments.

Line 135: Can this really be called an fsRIDL strategy? It makes females sterile but does not kill them, so the “L” isn't really applicable. A slight adjustment of terminology here may be appropriate.

Line 226-227: The change here helps, but it will also improve clarity to note that the X-shredding efficiency is imperfect, so the construct can occasionally end up in females, allowing for this cross to take place.

Line 337-347: I think this results section needs a caveat about the difficulty of generating MDFS combined with X-shredder. Please see my previous comment on Figures 6-7/Table 2. It's important to note that even after you get the combination, only half your offspring at best would have both, and throwing away half the offspring would reduce system efficiency in a different way. The additional to the discussion doesn't really describe this issue.

Line 390-391: Reference 75 doesn't seem to include homing (though still mentions HDR, just not in the context of gene drive). Also, I don't think that gRNA multiplexing has quite reached the level of “common practice”, even though it is more widespread than a few years ago.

Supplementary Figure 2: A: Are the samples from different developmental stages derived from separate generations or experimental batches? If so, this could introduce variability in cut rates. It seems only L4 was generated from MDFS males

crossed with females homozygous for a β 2tubulin-mCherry marker, while the other stages were not. Did these mosquitoes have the same genetic background? The figure seems to illustrate a trend of reaching maximum cut rate during larval development, but can this trend be reliably interpreted given that the samples may come from different genetic crosses?
B: Please clarify the meaning of the red triangles in the figure (insertions?)

References 85 and 86: Check for errors. For references in general, check that species' and genes' names are in italic.

Sincerely,
Jackson Champer

(Remarks on code availability)
code is okay

Reviewer #3

(Remarks to the Author)

(Remarks on code availability)

Response to reviewers

Reviewer #1 (Remarks to the Author):

This work from the Crisanti group provides an additional self-limiting genetic control for the control of malaria mosquitoes. This's been cleverly designed by using preexisting target sites and nuclease expressing constructs, i.e. *dsx* from Kyrou 2018 and X-lined rDNA from Galizi 2014, to develop a dominant female sterile approach. The transgenes are passed on at supermendelian frequencies exclusively via males, whilst females develop an intersex phenotype due to leaky expression of the CRISPR nuclease targeting the female-specific isoform of *dsx* or their proportion is further reduced in the case the X-targeting nuclease is also expressed during male meiosis.

The manuscript is well written and both experimental design and analysis/discussion look waterproof to me. I only have a few minor comments and suggestions noted below.

temporally and spatially restricted version of a homing gene drive 61 > The reference used doesn't seem the most appropriate here.

We appreciate the reviewer's comment and acknowledge that the reference used was inappropriate. We have replaced it with more suitable publications that describe homing drives.

the germline of both sexes and exhibits somatic leakiness 21,50,51.

> There might be other relevant works to be referenced here, particularly those looking at gene drive resistance.

We appreciate the reviewer's comment and have updated the references to include several more relevant publications that provide a detailed description of the features of the *vasa2* promoter.

vasa2 promoter would cause the expression of Cas9 in somatic cells.

> Has this been clearly confirmed vs deposition or only assumed?

We have previously demonstrated that the *vasa2* promoter results in high levels of end-joining repair in the embryo following maternal deposition of the endonuclease. Additionally, we reported that the expression of the endonuclease driven by this promoter outside of the mosquito germline—specifically, leaky expression in somatic tissues—caused significant unintended fitness costs in females carrying a single copy of the gene drive targeting a female-specific gene. This cost arose from the partial conversion of somatic cells to homozygosity for a null allele (Hammond et al., 2017 and 2021). In the context of the MDFS strategy, the transgene is always passed on to the offspring through males, and the deposition of endonucleases driven by the *vasa2* promoter has not been observed paternally. The dominant effect

of the MDFS in the female offspring is attributed to the leaky expression of nucleases in the somatic tissues.

two attP docking sites at the doublesex intron4-exon5 boundary (Kyrou et al., 2018; Supplementary Figure 1 and Figure 2A).

> It might be useful to briefly explain the difference vs similar iterations of this, e.g. Kyrou 2018 and Hammond 2016

We appreciate the reviewer's feedback and have revised the text to clarify how the MDFS construct was inserted at the *doublesex* locus. Furthermore, we have included a reference to Kyrou et al. (2018), which outlines a similar strategy for transgene integration at this locus.

subsequent molecular investigation confirmed the successful recombinase-mediated cassette exchange with the target locus (Supplementary Table 1).

> I don't see the molecular confirmation in Sup Tab 1, might be wrong reference?

We appreciate the reviewer's feedback. The incorrect reference to Supplementary Table 1 has been removed, and we have now included the PCR results confirming the integration of the transgene at the *doublesex* locus in Supplementary Figure 1.

Additionally, male accessory glands (MAGs) were present in all females, though in a small subset, they seemed less developed (e.g. there was only one or both were lighter in colour) (n=8/41) (Figure 3J).

> Is this the same phenotype shown in Kyrou 2018?

In the study conducted by Kyrou et al. (2018), an analysis of the internal reproductive organs of *dsxF*^{-/-} genotypic females revealed the presence of male accessory glands. This observation is similar to that found in MDFS females. This finding is illustrated in Supplementary Figure 2 of the Kyrou et al. (2018) study. We discuss the differences in the phenotype of these strains in paragraph 3 of the discussion.

that the penetrance of the dominant sterile phenotype, by virtue of the *vasa2* promoter, has remained stable over time.

> Data shown where/not shown?

We appreciate the reviewer's feedback. The MDFS strain has been maintained in our facility for over two years, and we regularly screen and monitor it as part of our routine activities. We have now included 'Supplementary Table 2,' which presents the transgenic rate observed in the MDFS strain's progeny and the occurrence of the intersex phenotype in MDFS females based on more than a year of monitoring.

This scenario allows the MDFS allele to transiently drive in the population, as it would be passed on by a greater than 50% portion of the progeny (i.e., the males).

> might be more useful earlier on in this section, e.g. near ref 43?

We appreciate the reviewer's comment. The transient driving dynamics discussed are a result of combining the MDFS construct with the X-shredder, rather than being due to the MDFS construct alone. We have revised the sentence for clarity and have integrated it more appropriately into the text.

were not statistically different to wild-type controls (77.6 ± 5.7 , $n=35$) ...and following fertility assays

> These differences are not significant due to low sample size - if that would be increased (e.g. as expected in field populations) and average N maintained the difference would be significant. This should be addressed in the text.

We appreciate the reviewers' comments and have revised the text to emphasise that this data refers specifically to comparing transgenic and wild-type males in laboratory settings.

The simulations included scenarios with an idealised...

> Please provide additional details, e.g. I don't see EJ in the ideal model?

We thank the reviewer for their feedback. We agree that more detail is needed, and we have now provided a supplementary methods section to describe the modelling in detail. The reviewer is correct in their observation that end joining is not included in the ideal model. This is because we set our cleavage parameter to 1 (100% of the target sites are cleaved) and end-joining parameter to 0 (0% end joining, 100% homing). This represents the idealised case where our homing rate is 100%, thus no end joining products are made. When the end-joining parameter is > 0 (as we have done for the simulations using parameters from the empirical data), the recessive NHEJ products are made and observed in the time series.

The supplementary methods should make this clearer, in particular Table 1.

This is largely due to the appearance of recessive NHEJ products, which confer a fitness advantage when compared to the dominant nature of the MDFS allele.

> What about fitness, would that have an impact too?

We did not observe any unintended fitness effects for the MDFS or the X-shredder strains, so the fitness parameters do not change between the empirical and optimal scenarios described in the modelling. Thus, the difference in the dynamics between

the optimal and empirical data is due to suboptimal cleavage, homing, and the presence of recessive NHEJ products. We have clarified this by including a supplementary methods section that provides a more detailed explanation of the parameters used for the modelling.

However, as with the single releases, recessive NHEJ products appear in the latter but not in the former (Figure 6B, D, F and H).

> See my previous comment about EJ, it would not be surprising if simply not included in the optimal model?

Yes, exactly. It is not included in the ideal model because homing is 100% (see previous comments and supplementary methods).

regardless of the extent of linkage between the two constructs (Table 1, Figure 7).

> Rates in the table and %s used in the figure do not help to compare the two. Possibly they could even be merged as unique figure

We appreciate the reviewer's comment. We agree that using %s in the text and release rates in the table and figure could cause some confusion. We have amended the text to quote the release rates directly and placed the percentage differences in parentheses.

(i.e., females are born hemizygous for the null allele), leading to an intersex phenotype that is less penetrant.

> what would be the impact of low or null penetrance hence fertile females? Would the transgene drive at that point?

We appreciate the reviewer's comment. Yes, if there is some residual fertility in heterozygote females there is a chance that the transgene could drive. We agree that this is an important question and have therefore performed some additional modelling and added some text to explain the conditions under which the construct is expected to drive: namely that drive is prevented if $h > 2 - 1/d$, where h is the fitness cost in heterozygous females and d is the transmission rate.

The cage trial allowed us to compare the in-vivo results of MDFS with those of other self-limiting strategies.

> is that the case? I could not find other constructs tested in the cage trial, maybe this should refer to the models?

Previously, we assessed the efficiency of an X-shredder in suppressing caged mosquito populations. Galizi et al. (2014) describe this cage trial in Figure 3. The reference to this work is provided in the text. The outcomes of these experiments

and the results of the MDFS cage trial align with the mathematical modelling predictions regarding the efficiency of both technologies.

Reviewer #2 (Remarks to the Author):

In this study, the authors make a dominant female-sterile homing drive in *Anopheles gambiae* for self-limiting population suppression. The drive works as expected, and mathematical modeling shows that this strategy is very promising for future field use.

Overall, I like the study, but there are several factors that reduce its impact:

1. The idea was proposed by the lab several years ago after an experimental demonstration, and these systems have also been modeled before.
2. A recent study (see below) made a similar drive in *A. stephensi*. It was a split drive, but had an important feature of two gRNAs.
3. This drive only has one gRNA. As indicated by a preprint that is referenced here, this means that the drive in this study can't be considered as a release candidate.

On the other hand, I appreciate the cage study, and the complete drive in a new species is also a useful advance. Thus, I support publication of the study in *Nature Communications* after some minor revisions.

Line 18: "Propose" may not be the best word to use in the context of this. As far as I know, the idea of male drive female sterile was originally proposed in Hammond 2016, and since then, two of our recent papers (Xu et al 2025 in *Nature Communications* and Han et al 2025 in *Molecular Biology & Evolution*) have elaborated on the idea quite a bit. Indeed, these two studies have several relevant features and should probably be incorporated throughout the manuscript (in particular, Xu et al made a split form of MSFD in *Anopheles* that had good performance).

We appreciate the reviewer's feedback and have made revisions to the text accordingly. We replaced the word "propose" with "describe" to clarify that we are not the first to present this strategy. Additionally, the recent papers highlighted by the reviewer have been incorporated into the manuscript (references 74 and 43).

Line 37: Should probably have a reference for this.

A reference has been added regarding the impact of insecticide resistance on malaria vector competence.

Line 53-55: This part could be made more concise.

We appreciate the reviewer's comment and have edited the text to be more concise.

Line 64: Not sure that "Contained" means in this sentence.

We appreciate the reviewer's comment and have edited the text accordingly.

Line 77-80: It's not just this. The gene drive will simply be more efficient at overall suppression (including suppressing the number of fertile females) if it targets a female-specific gene rather than a both-sex haplosufficient gene.

We appreciate the reviewer's comment and have edited the text accordingly.

Line 119: Reference 46 was a self-sustaining homing drive. Also, I'm not sure if the integral gene drive references here should necessarily be considered as self-limiting split drives, even though they do have a "split" aspect to them.

We appreciate the reviewer's comment and have removed the reference to the self-sustaining homing drive. However, we have kept the references to the integral gene drive, as we believe they provide valuable information and are relevant to the context (several aspects of these technologies have been assessed in their split version).

Line 128: "shredding" should probably be "shredder".

We appreciate the reviewer's comment and have edited the text accordingly.

Line 152: These two papers indicate that HDR may occur at fairly high rates (perhaps even the dominant form of repair) in flies, so this statement may be not quite right. Of course, for this drive, it doesn't matter whether HDR or end-joining occurs. Female mosquitoes would still be expected to become sterile.

Li Z, Marcel N, Devkota S, Auradkar A, Hedrick SM, Gantz VM, Bier E. CopyCatchers are versatile active genetic elements that detect and quantify inter-homolog somatic gene conversion. *Nat Commun*, 12, 1–12, 2021.

Du J, Chen W, Jia X, Xu X, Yang E, Zhou R, Zhang Y, Metzloff M, Messer PW, Champer J. Germline Cas9 promoters with improved performance for homing gene drive. *Nat Commun*, 15, 1–14, 2024.

Note that this affects the interpretation of Supplementary Figure 2 (it just shows the fraction of wild-type for alleles that have not been converted to drive).

We appreciate the reviewer's comment and acknowledge that our statement regarding repair mechanisms in the soma may not have been accurate. We have revised the text to address this concern and have updated the legend for Supplementary Figure 2. It now clarifies that the outcome of the sequencing results refers to the fraction of alleles that have been repaired through end joining.

Line 204: Why not show it if you have this data? This is an important target site, so knowing exactly what resistance alleles look like could be interesting.

We appreciate the reviewer's comments and have included representative examples of the sequencing data and repair outcomes observed for each analysed group in Supplementary Figure 2. Additionally, the raw amplicon sequencing data are available at the NCBI SRA under the accession code: PRJNA1227481.

Line 217-224: A brief word about getting the X-shredder in females would be useful here, for readers wondering how it would be possible (presumably imperfect efficiency means that it sometimes finds itself in females, which remain fertile due to male-specific expression).

We appreciate the reviewer's comment and have edited the text to increase clarity.

Line 291: Yadev et al. made a dominant-sterile drive targeting doublesex (like this manuscript), among other drives, but dominant-sterile resistance alleles were not identified.

We agree with the reviewer that the reference was not appropriate and have removed it from the text.

Line 347: "and fully" should be "and being fully".

We appreciate the reviewer's comment and have edited the text accordingly.

Line 367: "doublesex gene is highly functionally constrained" it would be more accurate to say the target site at dsx intron-exon boundary is constrained. Other target sites may not be as conserved and useful for gene drives.

We appreciate the reviewer's comment and have edited the text accordingly.

Line 374: Xu et al. (2025, Nature Communications) also showed this in *A. stephensi*.

We appreciate the reviewer's comment and have edited the text to include this reference.

Line 378: Another counterbalancing feature is that drive efficiency will likely go down with more than two gRNAs, though how much is unclear (S. Champer et al., Science

Advances 2020).

We appreciate the reviewer's comment, edited the text accordingly, and included the reference.

Line 420: References 57-58 (and your dominant-sterile resistance allele) may offer a solution - expressing the male isoform of dsx in females.

We appreciate the reviewer's comment; we have explored this possibility, and the results of the study will be included in a manuscript currently in preparation.

Line 430: "In vivo" should be italicized.

We appreciate the reviewer's comment and have edited the text accordingly.

Line 436-437: It's not good to reference an unpublished plasmid. If you used it in the manuscript, make the sequence available. At minimum, all the relevant annotated plasmid sequences in this study should be posted.

We appreciate the reviewer's comment and have included the reference to the plasmid sequence used, which has now been published.

Line 553-541: Can the numbers this was set to be justified in any way? It's okay if there is no firm reference, just saying something approximate would be sufficient.

We appreciate the reviewer's comment. The natural population growth rate referenced was derived from the work of Deredec et al. (2011). This rate was also used in the cage trial experiments described in Galizi et al. (2014). We have revised the text to include these citations.

Line 884: "Trail" should be "trial".

We appreciate the reviewer's comment and have edited the text accordingly.

References: Species names are not italic.

We appreciate the reviewer's comment and have edited the text accordingly.

Reference 57 has been published in Nature Communications.

We appreciate the reviewer's comment and the reference has now been updated.

Figure 2: The text in the image is a bit small.

We appreciate the reviewer's comment and have edited the text accordingly.

Figure 3: The labels E, F, G and I, H, J are reversed in the legend. Is the magnification the same for I, H, and J? Maybe make this clear in the legend.

We appreciate the reviewer's comment and have edited the text accordingly.

General considerations for Figure 6-7 and Table 2: These may be somewhat optimistic when considering practical combination of X-shredder and MDFS. It is often difficult to get an X-shredder in females. This limits your options for consistently generating homozygous individuals for release. In fact, a perfect X-shredder would make this even more difficult, requiring direct injection rather than lucky crosses. In this case and probably even for practical purposes for an imperfect X-shredder, you would probably still mostly be crossing males with a single drive and X-shredder copy to wild-type females in order to generate your release batch. Only half the male progeny will carry the X-shredder. You can keep using a few of these to stably generate the next generation, but your released males would still only have an X-shredder half the time unless you want to discard half your males. To avoid this, you'd need some sort of induction system. This could be mentioned, and it might also be useful to model releases where only half the males have X-shredders. It will still be helpful compared to just MDFS alone.

We appreciate the reviewer's comment and have included a note at the end of the discussion highlighting the need to develop strategies for large-scale rearing and sex sorting to potentially implement technologies such as the one described in this study in the field. Please note that for all MDFS simulations released males were heterozygous for the MDFS and X-shredder construct. When released in the same males, the MDFS and X-shredder were assumed to be on separate chromosomes.

In this study, I don't remember seeing anything about male mating success fitness. This wasn't measured. It's probably not a huge issue, but still worth mention that some additional fitness costs may somewhat slow down the drive.

We appreciate the reviewer's comment and have edited the text in the discussion to include this consideration.

Sincerely,

Jackson Champer

Reviewer #2 (Remarks on code availability):

Code is fine.

Reviewer #3 (Remarks to the Author):

Thank you

REVIEWER COMMENTS

Reviewer #1 (Remarks to the Author):

The authors have addressed my concerns and I don't have any other comments other than the two minor suggestions below - being only minor suggestions I don't need to see the revised manuscript again.

C1 vasa2 promoter would cause the expression of Cas9 in somatic cells.

> Has this been clearly confirmed vs deposition or only assumed?

R1 We have previously demonstrated that the vasa2 promoter results in high levels of end-joining repair in the embryo following maternal deposition of the endonuclease. Additionally, we reported that the expression of the endonuclease driven by this promoter outside of the mosquito germline—specifically, leaky expression in somatic tissues—caused significant unintended fitness costs in females carrying a single copy of the gene drive targeting a female-specific gene. This cost arose from the partial conversion of somatic cells to homozygosity for a null allele (Hammond et al., 2017 and 2021). In the context of the MDFS strategy, the transgene is always passed on to the offspring through males, and the deposition of endonucleases driven by the vasa2 promoter has not been observed paternally. The dominant effect of the MDFS in the female offspring is attributed to the leaky expression of nucleases in the somatic tissues.

C2 “The deposition of endonucleases driven by the vasa2 promoter has not been observed paternally” > It looks like this was not measured here and therefore the assumption may be derived from elsewhere? I would also assume that deposition would likely depend on dose of germline expression and nuclease thermostability. I still don't see how paternal deposition could be excluded here - unless provable otherwise I would recommend adding the paternal deposition as possible source/cause of somatic activity.

We apologise for omitting the reference supporting the statement that paternal deposition of endonucleases driven by the vasa2 promoter has not been observed. Hammond et al. (2017) developed a genetic screen to quantify parental nuclease deposition from various constructs, including those driven by the vasa2 promoter (Figure 3). They generated embryos carrying a wild-type target allele balanced against a functional, nuclease-resistant (r1) allele at a female fertility gene locus. These embryos, which did not carry a gene drive construct, were exposed to either paternal or maternal nuclease. In the absence of nuclease deposition, wild-type and r1 alleles are expected to appear in equal proportions, while novel indels indicate nuclease activity resulting from parental deposition. For vasa2-driven constructs, only maternal deposition was detected, affecting approximately 70% of nuclease-sensitive alleles, with no paternal deposition observed.

This is also reflected in the results from Tolosana et al. (2025), where daughters of males expressing a Y-linked vasa2-Cas9 and gRNA targeting the *dsx* locus, inherit a dominant negative allele (*dsxFΔ11*) at this locus—resulting in an intersex phenotype. The sequencing of these females revealed that approximately 50% of the reads corresponded to the *dsxFΔ11* allele and 50% to the wild-type *dsx* allele (Supplementary figure 1). This finding does not support the occurrence of *de novo* mutations from paternally deposited nuclease.

Additionally, Hammond et al. (2017) showed high transmission rates (>97%) when a vasa2-CRISPRh drive was inherited paternally. In contrast, maternal inheritance of the same gene drive caused a significant reduction in homing efficiency in males (~60.2%), likely due to nuclease deposition leading to the formation of resistant alleles in the embryonic germline. These results are shown in Figure 2B.

This is further supported by Tolosana et al. (2025) and our present study, where the transmission rates of the MDFS transgene—which is transmitted only paternally (as heterozygous females are sterile)—remain consistently high (~95%) across multiple subsequent generations (Supplementary Table 2).

C1 subsequent molecular investigation confirmed the successful recombinase-mediated cassette exchange with the target locus (Supplementary Table 1).

> I don't see the molecular confirmation in Sup Tab 1, might be wrong reference?

R1 We appreciate the reviewer's feedback. The incorrect reference to Supplementary Table 1 has been removed, and we have now included the PCR results confirming the integration of the transgene at the doublesex locus in Supplementary Figure 1.

C2 Might be useful to show the primers used in the schematic above to make the gel photo meaningful.

Supplementary Figure 1 has been revised to address the reviewer's request.

Reviewer #2 (Remarks to the Author):

The revisions are mostly quite good, and I think that the manuscript can be accepted now. We have a few more small comments.

Line 135: Can this really be called an fsRIDL strategy? It makes females sterile but does not kill them, so the "L" isn't really applicable. A slight adjustment of terminology here may be appropriate.

We appreciate the reviewer's concern. In Burt (2018), where the term "fs-RIDL-drive" was first introduced, the authors specifically used "fs-RIDL-drive-f" to describe a variant of this technology that induces sterility rather than lethality. However, the term fs-RIDL has since been applied more broadly to technologies inducing both lethality and sterility—for example, in Labbé et al. (2012), which describes a female-specific flightless (fsRIDL) phenotype used for controlling *Aedes albopictus*.

While we agree that the term RIDL is more precise for techniques inducing lethality, we believe that fs-RIDL can also appropriately describe sterility-inducing approaches, provided this distinction is clearly stated in the text. Alternatively, we could adopt the term fs-RIDL-drive-f as used in Burt (2018), though we fear this will reduce clarity or intuitiveness rather than improving on these.

Line 226-227: The change here helps, but it will also improve clarity to note that the X-shredding efficiency is imperfect, so the construct can occasionally end up in females, allowing for this cross to take place.

In the relevant section of the manuscript, we examine the fertility of males carrying the MDFS allele alone and males carrying the MDFS allele in combination with an unlinked X-shredder allele. While we appreciate the reviewer's insightful comment regarding the potential occurrence of rare escape females carrying the X-shredder construct (noting that Figure 4A depicts an idealised scenario), the presence or absence of such females does not impact the results presented in this section, as these analyses pertain exclusively to male fertility.

For clarity, we would like to specify that the X-shredder strain in our insectary is maintained through females rather than males. In females, the X-shredder allele is inactive. To maintain the strain, these females are routinely crossed with wild-type males, resulting in the X-shredder allele being inherited at approximately 50% frequency, equally distributed between male and female offspring. Consequently, setting up the cross between MDFS males and X-shredder females to generate MDFS; X-shredder trans-heterozygous males is straightforward, as females carrying the X-shredder allele are readily available and not a limiting factor.

Line 337-347: I think this results section needs a caveat about the difficulty of generating MDFS combined with X-shredder. Please see my previous comment on Figures 6-7/Table 2. It's important to note that even after you get the combination, only half your offspring at best would have both, and throwing away half the offspring would reduce system efficiency in a different way. The additional to the discussion doesn't really describe this issue.

We appreciate the reviewer's comment and understand the concerns regarding the challenges of generating MDFS combined with the X-shredder. We have addressed this point in the discussion by highlighting the current limitations and proposing potential strategies to improve the system.

Line 390-391: Reference 75 doesn't seem to include homing (though still mentions HDR, just not in the context of gene drive). Also, I don't think that gRNA multiplexing has quite reached the level of "common practice", even though it is more widespread than a few years ago.

We have replaced "common practice" with "becoming increasingly common" and removed reference 75 from the manuscript.

•

Supplementary Figure 2: A: Are the samples from different developmental stages derived from separate generations or experimental batches? If so, this could introduce variability in cut rates. It seems only L4 was generated from MDFS males crossed with females homozygous for a β 2tubulin-mCherry marker, while the other stages were not. Did these mosquitoes have the same genetic background? The figure seems to illustrate a trend of reaching maximum cut rate during larval development, but can this trend be reliably interpreted given that the samples may come from different genetic crosses?

We acknowledge the reviewer's concern. However, samples from different developmental stages were collected from replicate cages of MDFS males (G3 background) crossed with wild-type females (G3 background). Each analysed sample carries the MDFS construct (inherited from the fathers) and a wild-type allele (inherited from the mothers). The B2 Cherry marker was incorporated only in those individuals that were analysed at the L4 and pupal stages, and it is unlikely that the presence of this fluorescent marker would affect the observed trend in cut rates. Additionally, the reduced cleavage rates observed at target sites during the L4 and pupal stages relative to L1 were also maintained in adults, in which the B2 Cherry marker was not employed due to the ease of sex identification at this stage. This consistency supports the reliability of the cleavage rate trends reported.

B: Please clarify the meaning of the red triangles in the figure (insertions?)

Supplementary Figure 1 has been revised to address the reviewer's request.

References 85 and 86: Check for errors. For references in general, check that species' and genes' names are in italic.

The reference formatting has been revised.

Sincerely,
Jackson Champer

Reviewer #2 (Remarks on code availability):

code is okay

Reviewer #3 (Remarks to the Author):
